# Voriconazole Admixed with PMMA—Impact on Mechanical Properties and Efficacy

**DOI:** 10.3390/antibiotics12050848

**Published:** 2023-05-04

**Authors:** Barbara Krampitz, Julia Steiner, Andrej Trampuz, Klaus-Dieter Kühn

**Affiliations:** 1Medical Training and Science, Heraeus Medical GmhH, Hamburger Allee 50, 60486 Frankfurt, Germany; barbara.krampitz@t-online.de; 2University Hospital for Orthopaedics and Traumatology, Medical University of Graz, 8036 Graz, Austria; julia.christina.steiner@gmail.com; 3Infectious Diseases, Center for Musculoskeletal Surgery, Charité, Universitätsmedizin Berlin, 10117 Berlin, Germany; andrej.trampuz@charite.de

**Keywords:** voriconazole, PMMA cement, mechanical properties, efficacy, inhibition zone test, Candida spec

## Abstract

Background: There are currently no recommendations to direct the optimal diagnosis and treatment of fungal osteoarticular infections, including prosthetic joint infections and osteomyelitis. Active agents (fluconazole; amphotericin B) are regularly applied per os or intravenously. Other drugs such as voriconazole are used less frequently, especially locally. Voriconazole is less toxic and has promising results. Local antifungal medication during primary surgical treatment has been investigated by implanting an impregnated PMMA cement spacer using intra-articular powder or by daily intra-articular lavage. The admixed dosages are rarely based on characteristic values and microbiological and mechanical data. The purpose of this in vitro study is to investigate the mechanical stability and efficacy of antifungal-admixed PMMA with admixed voriconazole at low and high concentrations. Methods: Mechanical properties (ISO 5833 and DIN 53435) as well as efficacy with inhibition zone tests with two Candida spp. were investigated. We tested three separate cement bodies at each measuring time (*n* = 3) Results: Mixing high dosages of voriconazole causes white specks on inhomogeneous cement surfaces. ISO compression, ISO bending, and DIN impact were significantly reduced, and ISO bending modulus increased. There was a high efficacy against *C. albicans* with low and high voriconazole concentrations. Against *C. glabrata*, a high concentration of voriconazole was significantly more efficient than a dose at a low concentration. Conclusions: Mixing voriconazole powder with PMMA (Polymethylmethacrylate) powder homogeneously is not easy because of the high amount of dry voriconazole in the powder formulation. Adding voriconazole (a powder for infusion solutions) has a high impact on its mechanical properties. Efficacy is already good at low concentrations.

## 1. Introduction

Fungal periprosthetic joint infections (FPJIs) are considered a rarity [1,2,3]. These complications have a low incidence among PJIs [3,4,5,6]. Fungal infections are generally associated with bacteria. They affect patients with decreased immunity, especially during revisions [6,7] with polymicrobial infections [8,9]. They are most frequently identified as yeasts (especially Candida spp.), which are difficult to treat [10,11]. The antifungal agents most frequently admixed with PMMA are voriconazole [12], amphotericin B [13], fluconazole [14], and echinocandins [15]. The admixture of anti-infectives is a routine procedure for spacers and bone cement for fixation [16]. Endo Klinik uses antifungal agents in PMMA for one-stage septic fungal revisions. To combat fungal infection, attempting to use antifungal-loaded bone cement (ALFBC) in a spacer is promising during a recommended two-stage procedure [17,18]. The preparation and use of the drugs under sterile conditions is a prerequisite for admixture with PMMA. The use of pharmaceutical formulations requires a high amount of sterile powder with a small amount of pure voriconazole. This might have a significant influence on the mechanical properties, although small voriconazole dosages are admixed to PMMA [11,12,13]. Voriconazole has been selected because it is less nephrotoxic and has fewer side effects than amphotericin B [6,7,12]. The dose recommendations for voriconazole range from 200 to 600 mg/40 g PMMA cement [15,16] (Consensus Recommendations 2013, Pocket Guide). There are missing data in the literature concerning the ISO and DIN mechanical tests of voriconazole in the PMMA (ISO 5833:2002, DIN 53435:1983) [19,20].

To find suitable recommendations and optimized findings, the present study focuses upon the following questions: To what extent do admixed antifungals interfere with the mechanical stability of bone cement? Does voriconazole eluted from PMMA inhibit the growth of *Candida albicans* and *Candida glabrata* strongly in inhibition zone assays?

## 2. Results

The mixture of PMMA cement with a low dosage (200 mg) of voriconazole (44 g powder) was dry (Figure 1a,b). The production, especially with the high dosage of voriconazole, was distinctly dry with white specks. It was difficult to obtain a homogenous mixture.

Voriconazole was formed with the admixture of 10.5 g to the previously mentioned exception outside the ISO standard. The admixture of 3.5 g voriconazole also showed a reduced compressive strength but, at 74.4 MPa, it was still above the standard specifications. The vertical axis here describes the PMMA cement Palacos R + G used as a reference value (Figure 2).

### 2.1. ISO Bending Strength

A significant reduction in the bending strength could be observed after voriconazole was added to PMMA. Low dosages just met the ISO specification (50 MPa), and high dosages were well below it at 37 MPa (Figure 3).

### 2.2. ISO Bending Modulus

Admixing voriconazole with PMMA increased the ISO bending modulus with low dosages (2983 MPa), as well as with high concentrations (3040 MPa), in comparison to the reference (2769 MPa). The higher the dose of voriconazole in PMMA, the higher the bending modulus Figure 4.

### 2.3. DIN Impact Strength (Dynstat)

Dynstat impact strength was significantly reduced after the addition of voriconazole. In comparison to the reference (3.14 kJ/m^2^), the impact strength was made 43% lower by adding low dosages to PMMA (1.78 kJ/m^2^) and 64% after admixing high voriconazole concentrations (1.12 kJ/m^2^). The higher the voriconazole concentration in PMMA, the lowe r the DIN impact strength (Figure 5)

### 2.4. Inhibition Zone Tests

It was visible that voriconazole was active against *Candida albicans* and *Candida glabrata* over a period of 6 weeks after it was added to PMMA. The efficacy against *Candida albicans* was good at low and high dosages in PMMA. High voriconazole concentrations showed a better efficacy against *Candida albicans* up to day 15 compared with low dosages of voriconazole. (Figure 6. Low concentrations of voriconazole in PMMA against *Candida glabrata* were distinctly weaker than high dosages of voriconazole. Low concentrations of voriconazole showed a constant low efficacy against *Candida glabrata* over 42 d (Figure 7).

## 3. Materials and Methods

### 3.1. Materials

Palacos R + G was used as a PMMA reference. Appropriate amounts of voriconazole were added to Palacos R + G powder (Table 1). The voriconazole used was in the form of 200 mg powder (Rotexmedica), which was added manually to PMMA powder in low (200 mg) and high (600 mg) voriconazole dosages. The PMMA quantity of low dosages of voriconazole was 44 g and the quantity of high dosages was 51 g. The homogeneity of the powder and the set cement was visually detected.

Voriconazole, manufactured as a sterile powder for a solution for infusion, was delivered as a white-to-cream-colored lyophilized powder Hydroxy-Propyl Beta-Dex (HPBCD, Cyclodextrin) and was added within a range of 0.81–0.99 as a pharmaceutical additive in a vial. The holder of the marketing authorization is Xellia Pharmaceuticals, Kopenhagen, Denmark (Data Safety Data Sheet, source: URL http://www.cymitquimica.com/uploads/products/45/pdf/1329709-msds.pdf, accessed on 25 June 2020).

### 3.2. Methods

#### 3.2.1. Mechanical Testing

Compression strength was determined in MPa according to ISO 5833. A testing overview is summarized in Table 2. Cylindrical specimens with a diameter of 6 mm and a height of 12 mm were loaded with a constant crosshead speed of 19.8–25.4 mm/min. The tests were run at 23 ± 1 °C with dry specimens prepared 24 h before testing. According to ISO 5833 (2002) statistics, we calculated the average strength and standard deviation for 6 cylinders [19].

For ISO 5833 bending strength and bending modulus, rectangular specimens were used (3.3 × 75.0 × 10.0 mm). They were loaded with a constant crosshead speed of 5 mm/min. The tests were run at 23 ± 1 °C with dry specimens prepared 24 h before testing. The four-point test rig had 60 mm between the outer loading points and 20 mm between the inner loading points. The tests were continued until failure, and the maximum force was used to calculate the bending strength. The bending modulus was calculated from the difference between the deflections under loads of 15 N and 50 N [19].

According to ISO 5833 statistics, we calculated the average and the standard deviation of the values for six test specimens in MPa [19].

To determine the impact strength according to DIN 53435, 3 × 10 × 15 mm specimens were produced in special plates. The specimens were tested after storage for at least 12 h under standard climatic conditions using the appropriate impact direction (consumption of at least 10%, at most 80%, of the maximum impact by the test specimens). The test specimens needed to be placed exactly vertical in the test device. Then, the pendulum could be adjusted to 90° the height of the drop. The impact used is shown by the maximum indicator. Impact strength is often used as a parameter to describe the properties of materials being exposed to high and fast impacts, such as plastic materials for cars. Impact strength is a measure of the energy required to cause a material to fracture when struck by a sudden blow. According to DIN 53435 statistics, we calculated the average and the standard deviation of the values for 10 test specimens in kJ/m^2^ [20].

#### 3.2.2. Inhibition Zone Testing

For testing the efficacy of added voriconazole eluted from PMMA, we used the inhibition zone test. We used cylindric cement bodies of 25 mm diameter ×10 mm height. One cement body per dosage was incubated in vitro in a buffer solution at room temperature.

The voriconazole containing cement bodies were incubated in PBS buffer at room temperature. The measuring times were 1 h, 24 h, 7 d, 14 d, 28 d, and 42 d. After these measuring points as described before, a portion of this solution (60 µL = here called eluate) was taken and pipetted onto previously grown Candida colonies on Soubaroud plates (for 24 h at 37 °C). We tested 3 separate cement bodies at each measuring time (*n* = 3).The inhibition zone’s diameter was measured against *Candida albicans* and *Candida glabrata*. We calculated the average and the standard deviation of the zones of inhibition in mm (Figure 8a,b).

## 4. Discussion

We found that the mechanical stability (ISO compression, ISO bending, DIN impact strength) (Table 2) of bone cement decreases when voriconazole was admixed with one exception, the bending modulus, which increased.

Studies on the mechanical stability of voriconazole containing PMMA cement have been limited to compressive strength [2,12,21,22]. Reduced compressive strength was measured by Czuban et al. 2019 [22] after antifungal elution. According to this Czuban study, the reason for reduced compressive strength might be Gentamicin with Palacos R + G (industrially premixed with 0.5 g gentamicin), which was recognized as a poragen and was found to boost antifungal release [22,23]. Compressive strength slightly decreased, as the obtained values were above the level of the strength recommended for the implant fixation. Miller et al. (2013) [12] were the first to investigate compressive strength in antifungal-loaded Simplex P bone cement. The 600 mg voriconazole formulation was less than half of the required strength by the first day of elution. Miller et al. also noticed that the strength of both 300 mg and 600 mg was lower than recommended at the time for fixation.

According to our observations, the reduced compressive strength does not depend on the active ingredient release, but on the excessive powder quantity by admixing of voriconazole in PMMA (Recommendation: max 10% of the PMMA powder; actually, a low dose adds 7.9%, a high dose adds 26% of powder with voriconazole).

In ISO tests and after storage in a liquid medium, the question generally arises as to whether the ISO method can still be fulfilled and thus whether the limit values can be complied with. Our results show that high dosages of voriconazole mean 10.5 g powder (600 mg pure voriconazole) decreases the ISO slightly to 68 MPa compared to the ISO standard (70 MPa).

We did not find that ISO bending, DIN impact strength, and bending modulus were tested in other studies.

The bending strength of the high concentration of 600 mg voriconazole was significantly reduced to 37 MPa, with a limit of 50 MPa according to ISO. This means that the bending strength according to ISO is 26% below the limit value.

For DIN Impact strength, we found a significant reduction with low and high dosages of low voriconazole. It is well known that admixing [24,25] of antibioticswith PMMA influences mechanical properties. The more additional substance is added, the more it reduces the mechanical strength. The rule of thumb to add a maximum of 10% powder to the PMMA is confusing. The logic behind it is that the amount of active ingredient of voriconazole is not identical to the amount of powder. The commercial powder contains excipients (e.g., cyclodextrin) of up to 93.5%. If 200 mg of voriconazole is added, 3.5 g of a foreign substance is added to the PMMA (equal to plus 7.95%). If 600 mg of voriconazole is added, 10.5 g of the voriconazole powder substance is added, 26% instead of max. 10% is added to PMMA. From our point of view, the reduced mechanical solidity is only because it is not a pure substance but a powder for infusion. In addition to the active ingredient, they contain many other pharmaceutical excipients. Therefore, when adding voriconazole to PMMA cement, it is always necessary to carefully check which type of pharmaceutical formulation is used.

The mechanical properties of PMMA plus high/low voriconazole are reduced and, at the same time, PMMA plus low voriconazole is within the acceptable range of the ISO.

The manual admixture of high voriconazole dosages of 10.5 g (voriconazole powder for infusion contains 600 mg active ingredient) with PMMA increased the powder content to more than 10% of the total amount of manually prepared AFLBC. This significantly reduced the mechanical strength [26]. The most sensitive influence after admixing drugs with PMMA was the bending and impact strength [27].

The mechanical performance of PMMA cements is influenced by various parameters; as such, a larger amount of the substance increases the hydrophilicity [28,29]. The parameters are the composition of the cement, the porosity, and the preparation of the cement. The addition of radio opacifiers and antibiotics to bone cement slightly decreases its mechanical strength [26,30,31,32], as do antifungals such as voriconazole.

In addition, PMMA spacers have a relatively high mass and diameter, which are significantly more mechanically stable than thin cement layers for the fixation of the prosthesis. The DIN impact strength represents a stronger indication of mechanical sensitivity. Deviations from the standard spacers are investigated in solutions to approximate the in vivo situation.

Selected examples show that the results are currently not comparable (Table 3).

However, EUCAST susceptibility breakpoints have not been established due to a paucity of efficacy data. Due to the small number of patients with *C. glabrata* infections treated with voriconazole, clinical efficacy and pharmacokinetic (PK)–MIC data might help describe exposure [32].

In our in vitro study, voriconazole showed a very good effect against *Candida albicans* with inhibition peaks around 30 mm after a duration of 6 weeks. Against *Candida glabrata*, the inhibitory effect is borderline, especially when 300 mg was admixed. When admixing 600 mg, inhibition zones of 40 mm diameter were observed after 6 weeks, and there was thus a good inhibitory effect here.

WHO’s Fungal Priority List (FPPL) classifies *C. albicans* in the highest category as a Critical Priority Group and C. clabrata (Nakaseomyces glabrata) as a High-Priority Group. In 2022, the WHO FPPL made the first global effort to systematically prioritize fungal pathogens, considering their unmet R&D needs and perceived importance to public health [33]. Voriconazole is an essential medicine according to the WHO.

### 4.1. Considerations for Local Application of Voriconazole in PMMA

The local clinical application of voriconazole in spacers is intended to provide high-dose adjuvant antifungal therapy by releasing active substances at the site of infection. The dosage recommendations (0.3–0.6 g) [34] are rarely based on characteristic values, microbiological, or mechanical data. High doses of voriconazole are required to effectively control biofilm [13].

Sterile voriconazole is only available as a powder solution for infusion. These commercially available preparations contain a lot of pharmaceutical excipients in powder form but are low in the active ingredient voriconazole. Mixing high dosages of voriconazole causes white specks on the inhomogeneous cement surface. The visibility of the specks is particularly clear due to the green-colored cement, especially in high-dosage voriconazole (600 mg). Already, the mixing of the two powders (PMMA and voriconazole) reveals specks in an inhomogeneous mixture. It is recognizable at an early stage whether a mortar must be used for homogenization.

Studies do not normally specify the composition of voriconazole preparations, only the amount of the active ingredient in pure form is mentioned.

In the present study, voriconazole from the company Rotexmedica was used. Importantly, 200 mg and 600 mg of voriconazole (active ingredient) were provided in a powder for the infusion dosage form, with a total weight of 3.5 g and 10.5 g, respectively. The excipients unfavorably influence the mechanical strengths; moreover, they improve the release and efficacy. The efficacy of voriconazole is good even at low dosages.

The good efficacy, because poragen is included, is a disadvantage for the mechanical properties [28].

The exact compositions of the cement are often not specified.

### 4.2. Miscibility of Antifungals and Orthopedic Cement

Adjuvant therapy with local antimycotics in PMMA cement is becoming increasingly important [2]. Currently, no bone cement with admixed antimycotics is available commercially as an approved medical device. The surgeon must admix the substances manually as needed. The documented findings might support preparations of AFLBC onsite.

While mixing an AFLBC with voriconazole (as a powder for infusion), dose-dependent observations were witnessed.

First, the color and consistency changed when the two powders were mixed (PMMA and voriconazole). Depending on the dose of voriconazole added, the green cement powder used as a reference in our study became lighter in color. In addition, the miscibility became more difficult with increasing doses (see Figure 1b). This might be attributed to the lipophilicity of voriconazole [22,35].

The second observation was visible specks in the reference used. The green coloration made the contrast more visible. While mixing white voriconazole powder in white PMMA cement powder, inhomogeneities were hardly visible and may have appeared due to reduced hydrophilicity. This effect was particularly predominant at high dosages (600 mg in 10.5 g) of voriconazole (Figure 1d). To produce a homogeneous mixture, mortar is needed.

In the case of commercially available voriconazole formulations, the excipient ß-cyclodextrin accounts for 94% of the total weight.

Cyclodextrin is thus the limiting factor in voriconazole to increase the dosage. In addition, it impairs miscibility with orthopedic cement. The cement agglutinates or becomes crumbly. At the same time, ß-cyclodextrin improves the bioavailability of voriconazole.

Practical recommendations can be deduced to receive a homogenous cement dough:

Cool highly viscous orthopedic bone cement (powder and liquid monomer);

Use low viscosity orthopedic bone cement (especially for the spacer);

For spacer production extracorporeally, use a little more liquid monomer to obtain a homogeneous mixture/dough (p.r.n./if needed).

Cave: If more monomer liquid is used, the setting temperature in the residual monomer may increase.

Alternatively, only if sterile powdery antibiotics are not suitable or available for the admixture with PMMA, liquid antibiotics may be used as an effective alternative. If the admixture of liquid AB is inevitable, the admixture of liquid AB with the liquid monomer should be considered, followed by blending with the polymer powder. We discourage admixing liquid AB with PMMA powder or cement dough with low viscosity [36].

Studies have shown that the admixture of liquids with PMMA significantly reduces the mechanical properties and significantly deteriorates the effectiveness.

### 4.3. Future Considerations about Antifungal Dosage

It is questionable whether sterile voriconazole as a pure substance, if available, would show a better release from PMMA than voriconazole combined with pharmaceutical excipients. Of course, mechanical stability would hardly be affected by small amounts of voriconazole.

The release-supporting effect of currently available preparations for infusion could be omitted.

PMMA’s function as a drug carrier is currently being investigated. The incidence of fungal PJI is rising (Consensus 2018) [34,37,38] and has a much higher failure rate than bacterial infections, with eradication rates ranging between 50% and 93% [39], thus requiring further investigations, e.g., develop powdered voriconazole in pharmaceutical quality and galenic dosage formulation. Promising efforts include using PLGA (Poly-Lactic-co-Glycol Acid) to achieve a controlled antifungal release pattern. When added to PMMA or ALBC, it is biodegradable and has a release period in vitro ranging from weeks to months [39,40].

Investigations must elaborate on the ideal local antifungal concentration in the context of the standardized measurement of antifungal efficacy.

To effectively control biofilm [12], a high dosage is required. Here, based on the severity of the disease, the risk to the patient may be weighed up. The question of whether high local dosage justifies optional reduced mechanical stability remains to be investigated.

### 4.4. Limitations

Only one cement was used. It is an in vitro study, and no ethics committee would approve an in vivo study. We believe that an in vitro study can be used to collect data on a very good releasing bone cement. The data can only be reproduced for this one bone cement. Due to different compositions, different results can be expected for other bone cements.

## Figures and Tables

**Figure 1 antibiotics-12-00848-f001:**
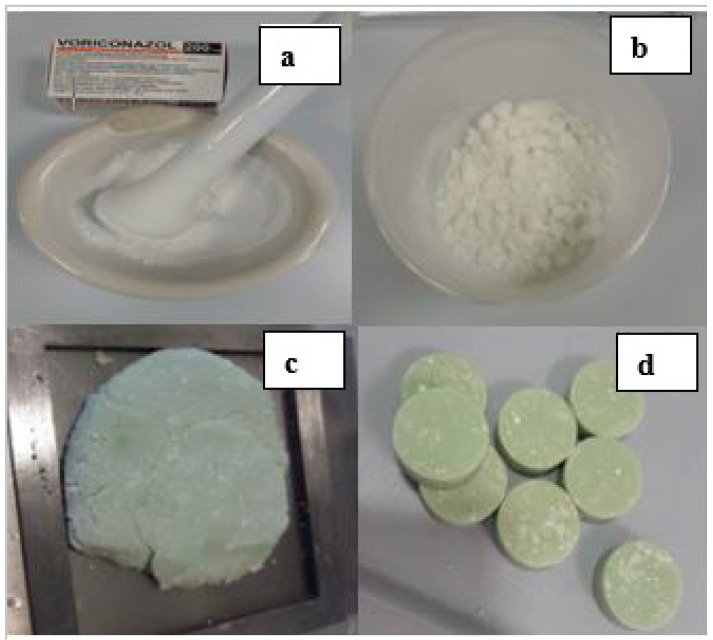
Samples of green PMMA with admixed voriconazole with white discoloration. (**a**): PMMA powder with low doses of voriconazole and mortar; (**b**): PMMA powder with a high dose of voriconazole. (**c**): PMMA dough with a low dose of voriconazole; (**d**): PMMA moldings with a high dose of voriconazole. (**a**–**d**): Difficulties in admixing voriconazole, especially dose-dependently.

**Figure 2 antibiotics-12-00848-f002:**
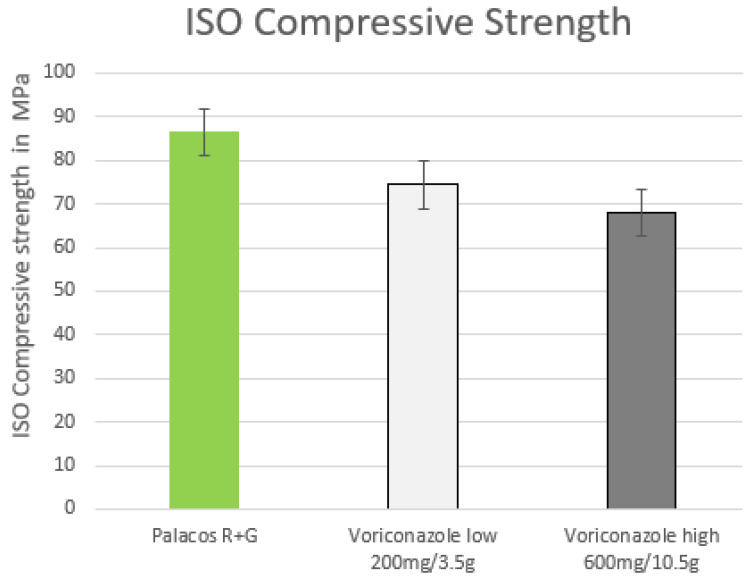
ISO compressive strength of PMMA cement samples with admixed low (200 mg) and high (600 mg) doses of voriconazole added to PMMA, measured in MPa (limit 70 MPa).The results of the compressive strength test corresponded to the standard specifications of ISO 5833, with one exception, which requires a limit value of at least 70 MPa. The exception was the addition of 10.5 g of voriconazole.

**Figure 3 antibiotics-12-00848-f003:**
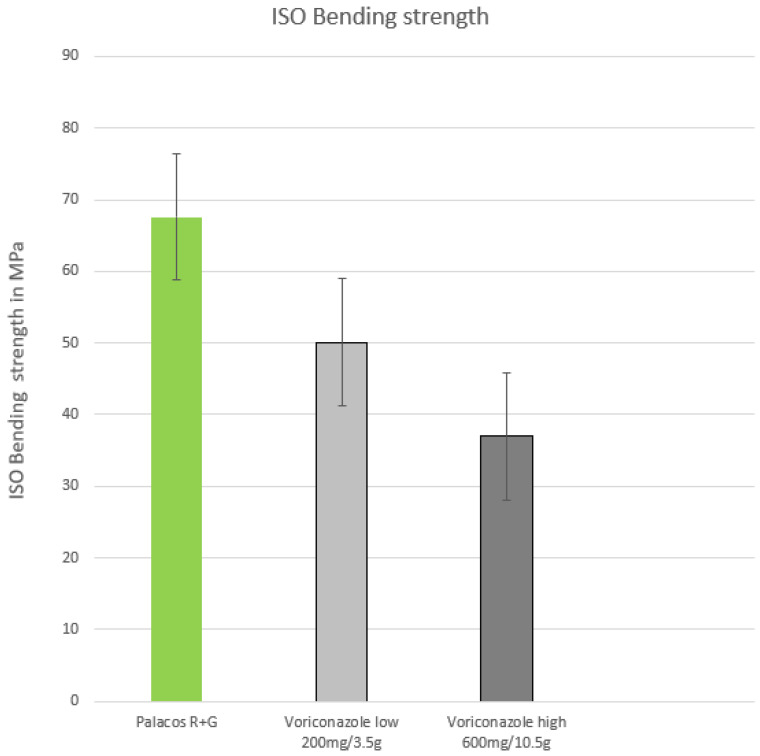
ISO bending strength of low (200 mg) and high (600 mg) voriconazole admixed with PMMA, measured in MPa (limit 50 MPa).

**Figure 4 antibiotics-12-00848-f004:**
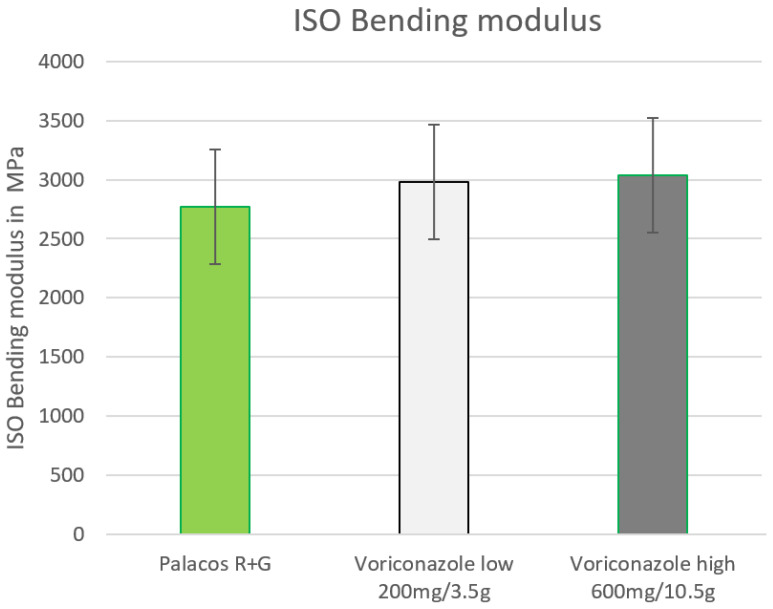
ISO bending modulus admixed low (200 mg) and high (600 mg) voriconazole to PMMA measured in MPa (limit 1800 MPa).

**Figure 5 antibiotics-12-00848-f005:**
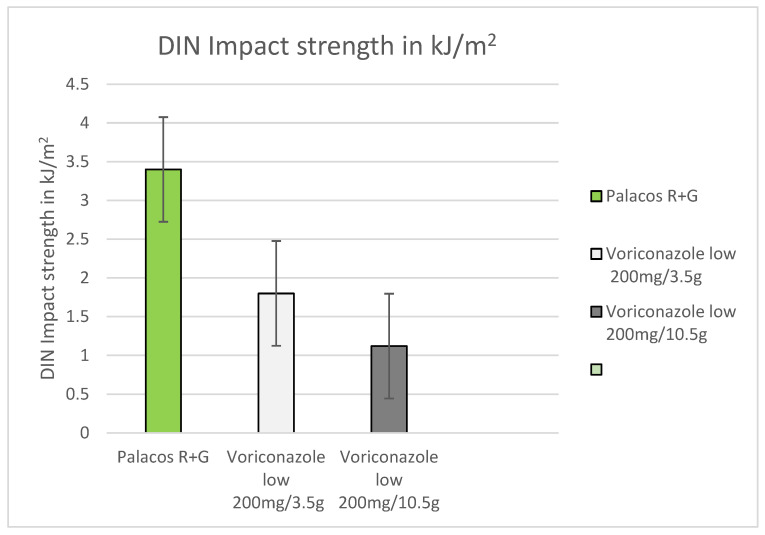
DIN impact strength with low (200 mg) and high (600 mg) levels of admixed voriconazole measured in kJ/m^2^ in comparison to PMMA without the addition of voriconazole.

**Figure 6 antibiotics-12-00848-f006:**
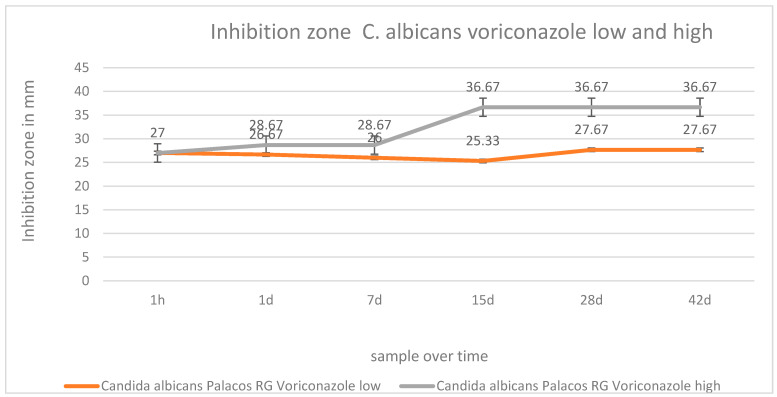
Inhibition zone test in mm of low and high concentrations of voriconazole in PMMA over 42 d against *Candida albicans*.

**Figure 7 antibiotics-12-00848-f007:**
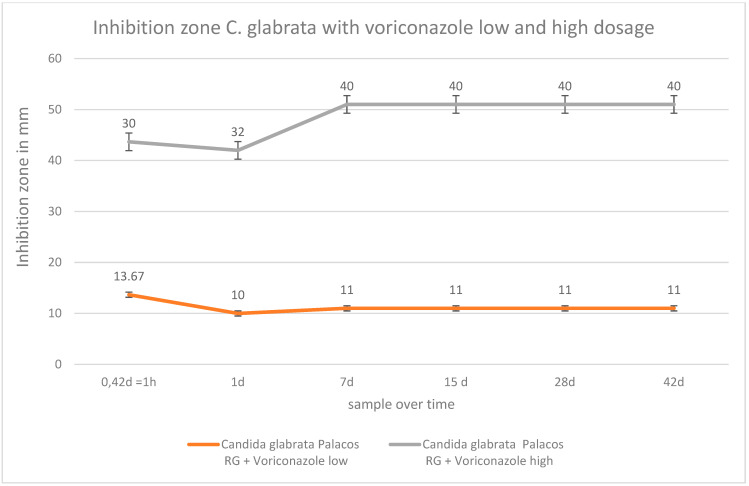
Inhibition zone test in mm of low and high concentrations of voriconazole in PMMA over 42 d against *Candida glabrata*.

**Figure 8 antibiotics-12-00848-f008:**
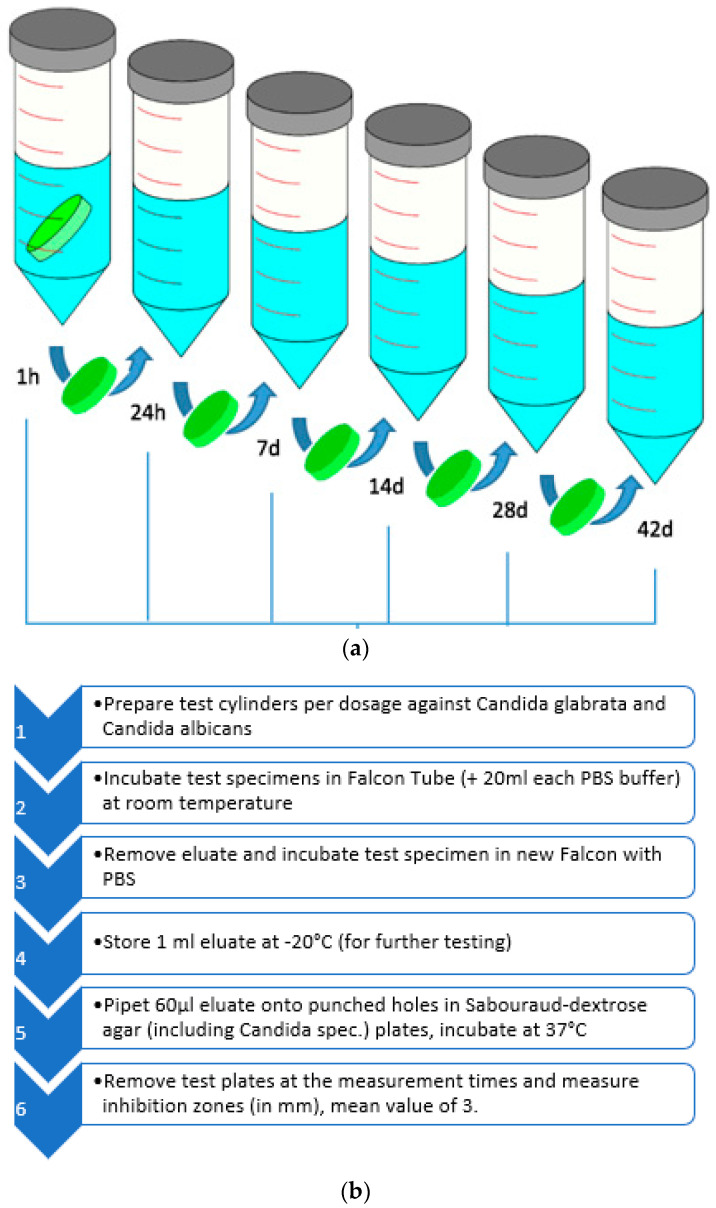
(**a**) The prepared moldings were incubated in PBS solution and transferred to a new container with fresh solution at each of the indicated times. The eluates thus obtained were used. (**b**) Overview of inhibition zone tests of voriconazole containing PMMA specimens with low and high dosages on Sabouraud agar with *Candida albicans* and *Candida glabrata*.

**Table 1 antibiotics-12-00848-t001:** Sample composition (g) and designation (mg) as well as powder amount after admixing voriconazole with PMMA powder (g).

	Pure Voriconazole in mg	Voriconazole (Rotexmedica) Powder for IV Solutions/in g (including Pharmaceutical Excipients)	Cement Powder Quantity after Adding Low and High Dosages of Voriconazole in g
40.5 g PMMA reference/sachet	-	-	40.5 g
40.5 g PMMA reference/sachet	+200 (low dose)	+3.5	44
40.5 g PMMA reference/sachet	+600 (high dose)	+10.5	51

**Table 2 antibiotics-12-00848-t002:** Testing overview: ISO 5833, DIN 53435 Inhibition Zone Test with high (600 mg) and low (200 mg) dosages of voriconazole. The dose–response and mechanical stability of the PMMA–voriconazole mixture were the target criteria. + tested/− not tested.

Test Series Overview	Voriconazole Low (200 mg) in PMMA Reference	Voriconazole High (600 mg) in PMMA Reference	PMMA Reference
Methods:			
ISO compression 70 MPa	+	+	+
ISO bending 50 MPa	+	+	+
ISO bending modulus			
1800 MPa	+	+	+
DIN impact strength (Dynstat)	+	+	+
Inhibition zone test	+	+	+
*C. albicans*	+	+	_
*C. glabrata*	+	+	_

**Table 3 antibiotics-12-00848-t003:** Comparison of commercially available voriconazole preparations.

Voriconazole	Dosage Form	Manufacturer/ Product Name	Voriconazole in mg Active Substance	Excipient in mg	Total Weight	The Active Substance with Respect to Total Weight	Publication Mentioned
Voriconazole	Powder for infusion	Hikma (before Hospira)/ Voriconazole 200 mg Hikma	200 mg	Each vial contains 217.6 mg of sodium. Each vial contains 3200 mg Sulphobutylether beta cyclodextrin sodium (SBECD) (powder for infusion).	3.417 mg	5.8% of the total weight is voriconazole. 93.5% of the total weight is SBECD.	Miller et al., 2013 [12]
Voriconazole	N/A	N/A	1000 mg	N/A	N/A	N/A	Deelstra et al., 2013 [32]
Voriconazole	Powder for infusion	Ratiopharm/ Voriconazole 200 mg Ratiopharm	200 mg	Hydroxypropylbetadex: 2500 mg	2.700 mg	7.4% of total weight is voriconazole. 92.6% of the total weight is hydroxypropylbetadex.	SmPC (Feb 2022)
Voriconazole	Powder for infusion	Rotexamedica	200 mg	Hydroxypropylbetadex (HPBCD): 3300 mg	3.500 mg	5.7% of total weight is voriconazole. 94.2% of the total weight is SBECD.	N/A
Voriconazole	Powder for infusion	Pfizer VFEND 200 mg	200 mg	Natrium: 221 mg Beta-cyclodextrin-sulfobutylether (SBECD): 3200 mg	3.621 mg	5.5% of the total weight is voriconazole. 94% of the total weight is SBECD.	SmPC (May 2022)

N/A = not available or no answer.

## Data Availability

All data are presented in this article. The data are also available on the Medical University of Graz (MUG).

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
