# Peer review of "Voriconazole Admixed with PMMA—Impact on Mechanical Properties and Efficacy"

_antibiotics, 2023, doi:10.3390/antibiotics12050848_

Round 1

Reviewer 1 Report

there are some inaccuracies, possibly lack of clarity due to languae use which require attention.

Line 24: white tapping-- is there a better word in English to describe what you mean?

Line 38: you cite 3% frequency for fungal PJI. Your  references  support support the commonly acccepted rate of 1-3% total PJI  rate(vs. total arthroplasties performed) , of which fungal PJI is cited as very rare; by my estimation fungal PJI based on this would have a net rate far below 1% of total joint replacements. Do you mean fungal represents 3% of all PJI? This needs to be clarified and supported by other references.

line 44: typo "isless" >> "is less" 

I cannot figure out how you performed your efficacy testing. Were the PMMA/ antifungal cylinders incubated in buffer for each of the time durations of 1 hour to 42 days? Was the same eluate left in situ for a total of 42 days, accumulating eluted antibiotic continually? Or was the cylinder rinsed and re-incubated every day, noting the decreased elution over time?  was there a preexisting fungal lawn on the plates receiving the pipetted eluate? Or was a solution of fungus introduced to the plate of eluate? was there any repetition or multiple test samples at any point? I diagram or photo of the microbiological testing might be helpful, along with further description.

line 473-475 table formatting may need adjustment

the mechanical testing discusion was clear

was any efficacy testing done for the various forms of amphotericin?  I am not sure why the ampho b discussion was included in this paper. There does not seem to be parallel treatment with voriconazole.  mechanical testing of  amphotericin B  in various forms is loosely discussed but not compared directly in efficacy or strength. 

Author Response

To answer point by point:

Response to Editor:

Dear Editor,

we would like to thank you very much for your critical review and suggestions for improvement. Please see the note on how we plan to implement your improvements and whether this meets the essence of your comments.

Comments and Suggestions for Authors

Point 1 :there are some inaccuracies, possibly lack of clarity due to language use which require attention.

Line 24: white tapping-- is there a better word in English to describe what you mean? 

Response to Editor Point 1:  Yes, we exchange the expressions. We removed terms like specks to white tapping. “Mixing high dosages of voriconazole causes white specks tapping on inhomogeneous cement” surfaces

Point 2 Line 38: you cite 3% frequency for fungal PJI. Your references support support the commonly acccepted rate of 1-3% totalPJI rate(vs. total arthroplasties performed) , of which fungal PJIis cited as very rare; by my estimation fungal PJI based on thiswould have a net rate far below 1% of total joint replacements.Do you mean fungal represents 3% of all PJI? This needs to beclarified and supported by other references.

Point 2 Response to Editor: Fungal periprosthetic joint infections (PJIs) are considered a rarity, with approximately 3% frequency of all PJI [1,2,3]. These complications have a low incidence among PJIs [3,4,5,6]. “Fungal periprosthetic joint infections (FPJIs) are considered a rarity [1,2,3]. These complications have a low incidence among PJIs [3,4,5,6]. Fungal infections are generally associated with bacteria. They affect patients with decreased immunity, especially during revisions [6,7] with polymicrobial infections [8.9].”

Point 3 line 44: typo "isless" >> "is less"

Point 3 Response to Editor: typo corrected

Point 4 I cannot figure out how you performed your efficacy testing.
Point 4 Response to EditorFor testing the efficacy of added voriconazole eluted from PMMA we used inhibition zone test. In this process one cement body per dosage was incubated in-vitro in a buffer solution at room temperature

Point 4 Response to Editor:
New Figure 2a : Elution method

Explanatory Text:  The prepared moldings were incubated in PBS solution and transferred to a new container with fresh solution at each of the indicated times. The eluates thus obtained were used.

Point 5 Were the PMMA/ antifungal cylinders incubated in buffer for each of the time durations of 1 hour to 42 days?
Point 5 Response to the editor:  For that we used cylindric cement bodies 25 mm diameter x10 mm height. The voriconazole containing cement bodies were incubated in PBS buffer at room temperature. The measuring times were 1h, 24h, 7d, 14d, 28d, and 42d. After this measuring points as described before, a portion of this solution (60µl = here called eluate) was taken and pipetted onto previously grown Candida colonies on Soubaroud plates (for 24h at 37°C).

Point 6 Was the same eluate left in situ for a total of 42 days, accumulating eluted antibiotic continually?
Point 6
New Figure 2: Elution method, please see above

Or was the cylinder rinsed and re-incubated every day, noting the decreased elution over time?
New Figure 2: Elution method, please see above

Point 7 Was there a preexisting fungal lawn on the plates receiving the pipetted eluate?

Point 7 Response to the Editor: After this measuring points as described before, a portion of this solution (60µl = here called eluate) was taken and pipetted onto previously grown Candida colonies on Soubaroud plates (for 24h at 37°C). The inhibition zone’s diameter was measured against Candida albicans and Candida glabrata. We calculated the average and the standard deviation of the zones of inhibition in mm (Fig. 2a, Fig 2b).

Point 8 Or was a solution of fungus introduced to the plate of eluate?
Point 8 Response to Editor: Plates were prepared please see above

Point 9 Was there any repetition or multiple test sample set any point?

Point 9 Response to Editor: Plates were prepared
Always 3 separate specimens were investigated (n=3)
Point 9 We tested 3 separate cement bodies at each measuring time (n=3).

Point 10 A diagram or photo of the microbiological testing might be helpful, along with further description.
Point 10 Response to the editor: please see the Fig. 2a  included above.

Point 11 line 473-475 table formatting may need adjustment
Point 11 => Response to Editor:  tables are formatted

Point 12 the mechanical testing discusion was clear  
Point 12 Response to Editor:  Fine, so there is nothing to change

Point 13Was any efficacy testing done for the various forms of amphotericin?
Point 13 Response to the Editor:  The chapter is deleted.

Point 14 I am not sure why the ampho b discussion was included in this paper. There does not seem to be parallel treatment with voriconazole. mechanical testing of amphotericin B in various forms is loosely discussed but not compare directly in efficacy or strength.

Point 14 Response to Editor:  
Point 14 Response to the Editor: The chapter is deleted.

We have mentioned amphotericin B exclusively in the discussion.

The enquiries we receive at the MUG (Graz, Austria) and the Charité (Berlin) almost exclusively concern the influence of amphotericin or voriconazole on the mechanical properties on PMMA bone cement. That’s what the information seemed for us an added value to the practice of orthopedic surgeons.

Reviewer 2 Report

This is an very interesting paper concerning the mechanical properties of PMMA with voriconazole and the its antifungial effect.

The quality of the presentation must be improved significantly. The way of presentation is very confusing and I cannot believe that the last author did read that paper.

General comments:

ABSTRACT: The description of the methods and results in the abstract must be specific.

INTRODUCTION: Describe the value of fungal PJI. 3 % is wrong, maybe 3 % of all PJI. Than describe that mixing antibiotics in PMMA is a well known technique in treating PJI. Desribe the clinical effectivity of mixing antifungeal drugs in PMMA. Than describe the advantage of voriconazole and that missing data are in the literature concerning the mechanical tests of voriconazole in the PMMA. Than the aim and questions of the study

METHODS: Literature research can be deleted.

DISCUSSION: Must completely be rewritten. Repete shortly your results and compare these with what is known from the literature. Recommandations and statements like line 285 and the following lines must be supported by citations from the literature. Delete the paragraphs concerning Amphotericine. This is not topic of your paper. Discuss limitations of the study at the end before the  conclusions.

Author Response

To answer point by point:

Dear Editor,

we would like to thank you very much for your critical review and suggestions for improvement. Please see the note on how we plan to implement your improvements and whether this meets the essence of your comments.

Comments and Suggestions for Authors

This is an very interesting paper concerning the mechanicalproperties of PMMA with voriconazole and the its antifungialeffect.

The quality of the presentation must be improved significantly.The way of presentation is very confusing and I cannot believethat the last author did read that paper.

Point 1 General comments:

ABSTRACT: The description of the methods and results in theabstract must be specific.
Answer Point 1
Methods
: Mechanical properties (ISO 5833 and DIN 53435), as well as inhibition zone tests with two Candida spp., were investigated. Results: Sterile voriconazole is only available pulverized as a powder for solution (for infusion). These commercially available preparations contain a lot of pharmaceutical excipients in powder form but are low in the active ingredient voriconazole. Mixing high dosages of voriconazole causes white specks tapping on inhomogeneous cement surfaces. ISO compression, ISO bending, and DIN impact were significantly reduced, and ISO bending modulus increased. There was a high efficacy against C. albicans with low and high voriconazole concentrations. Against C. glabrata, a high concentration of voriconazole was significantly more efficient than a dose at a low concentration.

Point 2 INTRODUCTION: Describe the value of fungal PJI. 3 % iswrong, maybe 3 % of all PJI. 3% of all PJI

Than describe that mixingantibiotics in PMMA is a well known technique in treating PJI.

Point 2 Answer: Admixture of anti-infectives is a routine procedure for spacers and bone cement for fixation.

Point 3 Then describe the advantage of voriconazole and that missing data are in the literature concerning the mechanical tests of voriconazole in the PMMA.
Point 3 Answer There are no data

Point 4 Then the aim and questions of the study

Point 4 Answer: The text is restructured according to you suggestion

  1. Introduction

Fungal periprosthetic joint infections (FPJIs) are considered a rarity [1,2,3].  These complications have a low incidence among PJIs [3,4,5,6]. Fungal infections are generally associated with bacteria. They affect patients with decreased immunity, especially during revisions [6,7] with polymicrobial infections [8.9]. They are most frequently identified as yeasts (especially Candida spp.), which are difficult to treat [10,11]. The antifungal agents most frequently admixed with PMMA are voriconazole [12], amphotericin B [13], fluconazole [14], and echinocandins [15].  Admixture of anti-infectives is a routine procedure for spacers and bone cement for fixation . To combat fungal infection, attempting to use antifungal-loaded bone cement (ALFBC) in a spacer is promising during a recommended two-stage procedure [17,18]. The preparation and use of the drugs under sterile conditions is a prerequisite for admixture with PMMA. The use of pharmaceutical formulations requires a high amount of sterile powder with a small amount of pure voriconazole. This might have a significant influence on the mechanical properties, although small voriconazole dosages are admixed to PMMA [11,12,13]. Voriconazole has been selected because it is less  (nephro-) toxic and has fewer side effects than amphotericin B compared to amphotericin B [6,7,12]. The dose recommendations for voriconazole range from 200 to 600 mg/40g PMMA Cement [15, 16] (Consensus Recommendations 2013; Pocket Guide). There are missing data in the literature concerning the ISO and DIN mechanical tests of voriconazole in the PMMA (ISO 5833:2002, DIN 53435:1983) (17,18)

To find suitable recommendations and optimized findings, the present study focuses upon the following questions: To what extend do admixed antifungals interfere with the mechanical stability of bone cement? Does voriconazole eluted from PMMA inhibit the growth of Candida albicans and Candida glabrata strongly in inhibition zone assays ?

Point 5 METHODS: Literature research can be deleted.

 Point 5 Answer: Literature research is deleted

Point 6 DISCUSSION: Must completely be rewritten.
Point 6 Answer: The discussion is completely rewritten.

Repeat shortly your results and compare these with what is known from the literature.

  1. Discussion

    We found mechanical stability (ISO compression, ISO bending, DIN impact strength) (Tab.2) of bone cement decreases when voriconazole was admixed, with one exception, the bending modulus, which increased.
    Studies on mechanical stability of voriconazole containing PMMA cement have been limited to compressive strength [2,12,30,31]. Reduced compressive strength was measured by Czuban et al. 2019 [31] after antifungal elution. According to this Czuban study the reason for reduced compressive strength might be Gentamicin with Palacos R+G (industrially premixed with 0.5 g gentamicin), was recognized as a poragen and was found to boost antifungal release [31,32]. Compressive strength slightly decreased, as the obtained values were above the level of the strength recommended for the implant fixation. Miller et al. (2013) [12] were the first to investigate compressive strength in antifungal-loaded Simplex P bone cement. The 600mg voriconazole formulation was less than half of the required strength by the first day of elution. Miller et al. also noticed that the strength of both 300mg and 600mg was lower than recommended at the time for fixation.

    According to our observations, the reduced compressive strength does not depend on the active ingredient release, but with the excessive powder quantity by admixing of voriconazole in PMMA (Recommendation: max 10 % of the PMMA powder; actually, low dose adds 7,9 %, high dose adds 26 % of powder with voriconazole).

In ISO tests and after storage in a liquid medium, the question generally arises as to whether the ISO method can still be fulfilled and thus whether the limit values can be complied with. Our results show that high dosages of voriconazole means 10,5g powder (600mg pure voriconazole) decreases the ISO slightly to 68 MPa compared to the ISO standard (70 MPa ).

However, Fink et al. (2011) [33] proved that, using the spacer technique 6 weeks after spacer implantation, the concentrations of antibiotic are sufficient to treat a periprosthetic infection.

We did not find ISO bending, DIN impact strength and bending modulus were tested in    other studies.

The bending strength of the high concentration of 600mg voriconazole was significantly reduced to 37MPa with a limit of 50 MPa according to ISO. This means that the bending strength according to ISO is 26% below the limit .

DIN Impact strength (Dynstat)
For DIN Impact strength we found significant reduction with low and high dosages of low voriconazole. It ist well known that the admixture (37, 51) of antibiotics and influences mechanical properties. The more additional substance is added the more it reduces the mechanical strength. The rule of thumb to add a maximum of 10% powder to the PMMA is confusing. The logic behing is that the amount of active ingredient of voriconazole is not identical to the amount of powder. The commercial powder contains excipients (e.g. cyclodextrin) of up to 93.5%. If 200 mg of voriconazole is added, 3.5 g of a foreign substance is added to the PMMA (equal to plus 7.95%). If 600 mg of voriconazole is added, 10.5 g of the voriconazole powder substance is added, 26% instead of max. 10% is added to PMMA. From our point of view, the reduced mechanical solidity is only because it is not a pure substance but a powder for infusion. In addition to the active ingredient, they contain many other pharmaceutical excipients.Therefore, when adding voriconazole to PMMA cement, it is always necessary to carefully check which type of pharmaceutical formulation is used.

The mechanical properties with PMMA plus voriconazole high/low are reduced and at the same time, PMMA plus low voriconazole is within the acceptable range of the ISO.                                                                                                                                                                                                                                                                                                                                                                                                                                                                                                                                                                                                                                                                                                       

The manual admixture of high voriconazole dosages of 10.5g (voriconazole powder for infusion contains 600mg active ingredient) with PMMA increased the powder content to more than 10% of the total amount of manually prepared AFLBC. This significantly reduced the mechanical strength [33]. The most sensitive influence after admixing drugs with PMMA was the bending and impact strength [35].

The mechanical performance of PMMA cements is influenced by various parameters; as such, a larger amount of the substance increases the hydrophilicity [30,31]. The parameters are the composition of the cement, the porosity, and the preparation of the cement. The addition of radio opacifiers and antibiotics to bone cement slightly decreases its mechanical strength [32,33], as do antifungals such as voriconazole.

In addition, PMMA spacers have a relatively high mass and diameter, which are significantly more mechanically stable than thin cement layers for the fixation of the prosthesis. DIN impact strength represents a stronger indication of mechanical sensitivity. Deviations from the standard spacers are investigated in solutions to approximate the in vivo situation.

Selected examples show that results are currently not comparable.

Point 7 Recommendations and statements like line 285 and the following lines must be supported by citations from the literature. Text line 285 “In addition, it impairs miscibility with orthopedic cement. The cement agglutinates or becomes crumbly. At the same time, ß-cyclodextrin improves the bioavailability of voriconazole…”

Point 7 Answer:: We would like to quote some literature. At the same time it has not been published anywhere. It has never been described before.

We are the first to publish it. The results are from our investigations

Point 8 Delete the paragraphs concerning Amphotericin. This is not topic of your paper.
Point 8 Answer: The information about Amphotericin is deleted.

Point 9: Discuss limitations of the study at the end before the conclusions.
Point 9 Authors Answer: Limitations: only one cement was used.

It is an in vitro study and no ethics committee would approve an in vivo study. We believe that an invitro study can be used to collect data on a very good releasing bone cement.  The data can only be reproduced for this one bone cement. Due to different composition, different results can be expected for other bone cements.

Limitations

Only one cement was used

It is an in vitro study, and no ethics committee would approve an in vivo study. We believe that an in vitro study can be used to collect data on a very good releasing bone cement.  The data can only be reproduced for this one bone cement. Due to different composition, different results can be expected for other bone cements .

Editor I Line 38: you cite 3% frequency for fungal PJI. Your references support support the commonly acccepted rate of 1-3% totalPJI rate(vs. total arthroplasties performed) , of which fungal PJIis cited as very rare; by my estimation fungal PJI based on thiswould have a net rate far below 1% of total joint replacements.Do you mean fungal represents 3% of all PJI? This needs to beclarified and supported by other references.

Edtor II INTRODUCTION: Describe the value of fungal PJI. 3 % iswrong, maybe 3 % of all PJI. 3% of all PJI
Response to Editors: Fungal periprosthetic joint infections (PJIs) are considered a rarity, with approximately 3% frequency of all PJI [1,2,3]. These complications have a low incidence among PJIs [3,4,5,6].

Editor II: Than describe that mixingantibiotics in PMMA is a well known technique in treating PJI.

Response to Editor II Admixture of anti-infectives is a routine procedure for spacers and bone cement for fixation.(Sentence included)

Editor I line 44: typo "isless" >> "is less"
Response to Editor I  typo is corrected

Editor II Than describe that mixingantibiotics in PMMA is a well known technique in treating PJI.

Answer: Admixture of anti-infectives is a routine procedure for spacers and bone cement for fixation.

Then describe the advantage of voriconazole and that missing data are in the literature concerning the mechanical tests of voriconazole in the PMMA.
Answer There are no data

Then the aim and questions of the study

Answer: The text is restructured according to you suggestion

Editor II Discuss limitations of the study at the end before the conclusions.
Authors Answer: :Limitations: only one cement was used.

Round 2

Reviewer 2 Report

The revised version is improved. There are only view remarks.

Abstract: Methods: Mention how many specimens were used.

               Results: Delete the first two sentences.

Introduction: Line 43: Mention that Endoklinik using antifungeal agents in PMMA for one-stage septic fungal revisions.

Line 44: Add references for using antifungeal agents in spacers.

Line: 51: delete "compared to amphotericin B"

Material and Methods: Delete Fig. 1

Discussion: Delete Line 216 to 218 or transpose it to the another place in the introduction (at the beginning) or discussion  where it fits better in the context.

Author Response

Dear Reviewer 2
Thank you for reviewing once more and signing the manuscript of “Voriconazole admixed with PMMA- Impact on mechanical properties and efficacy”.
We would now like to refer to the minor concerns and edit it according to your suggestions. Thank you for corrections.

Reviewer 2 Comment Abstract: Methods: Mention how many specimens were used.
Authors Answer We tested 3 separate cement bodies at each measuring time (n=3)

Reviewer 2 Comment Results: Delete the first two sentences.
Authors Answer  The first two sentences are deleted

Reviewer 2 Comment Introduction: Line 43: Mention that Endoklinik using antifungeal agents in PMMA for one-stage septic fungal revisions.
Authors Answer mentioned Endoklinik using antifungeal agents in PMMA for one-stage septic fungal revisions.

Authors Answer: The sentence is included. Endoklinik using antifungeal agents in PMMA for one-stage septic fungal revisions.

Reviewer 2 Comment Line 44: Add references for using antifungeal agents in spacers.

Authors Answer Author included: Dennes E.;Fiorenza F. Saint-Marcoux F.; Megherbi M.; Dupon M., Weinbreck P. Voriconazole stability in cement spacers. Med Mal Infect. 2012 Nov;42(11):567-8. doi: 10.1016/j.medmal.2012.07.007. Epub 2012 Oct 6. PMID: 23044087.

Reviewer 2 Comment Line: 51: delete "compared to amphotericin B"
Authors Answer: is deleted

Reviewer 2 Comment Material and Methods: Delete Fig. 1

Reviewer 2 Comment Discussion:
Delete Line 216 to 218 or transpose it to the another place in the introduction (at the beginning) or discussion  where it fits better in the context.

Authors : Due to a shift of pages we could not identify which sentences are meant. Would you mind giving us some more information.
